# Glycosylation of Immune Receptors in Cancer

**DOI:** 10.3390/cells10051100

**Published:** 2021-05-04

**Authors:** Ruoxuan Sun, Alyssa Min Jung Kim, Seung-Oe Lim

**Affiliations:** Department of Medicinal Chemistry and Molecular Pharmacology, Purdue Institute of Drug Discovery, Purdue Center for Cancer Research, Purdue University, West Lafayette, IN 47907, USA; sun714@purdue.edu (R.S.); kim1705@purdue.edu (A.M.J.K.)

**Keywords:** glycosylation, N-glycan, immune receptor, immune checkpoint therapy, cancer

## Abstract

Evading host immune surveillance is one of the hallmarks of cancer. Immune checkpoint therapy, which aims to eliminate cancer progression by reprogramming the antitumor immune response, currently occupies a solid position in the rapidly expanding arsenal of cancer therapy. As most immune checkpoints are membrane glycoproteins, mounting attention is drawn to asking how protein glycosylation affects immune function. The answers to this fundamental question will stimulate the rational development of future cancer diagnostics and therapeutic strategies.

## 1. Introduction

The immune system can eliminate malignancy at an early stage by recognizing the antigenic peptide epitope presented by neoplastically transformed cells [1,2]. However, a subset of tumor cells may evolve to coexist with antitumor immunity by a process defined as immunoediting [3]. To combat the immune attack and maintain the balance with the host, cancer cells develop a series of approaches such as (1) downregulation of antigen presentation to make them invisible to immunity/the immune system, (2) production of immunomodulatory cytokines to establish an immune suppressive environment, and (3) expression of surface immune checkpoint proteins such as programmed cell death 1 ligand 1 (PD-L1) that engages with its receptor, programmed cell death protein 1 (PD-1), on immune cells to inhibit the T-cell-mediated immune response. Restoring antitumor immunity by targeting suppressive or stimulatory immune checkpoints has resulted in immense achievements and greatly revolutionized the landscape of cancer therapy.

Glycosylation, which is defined as the association of a carbohydrate moiety to the substrate protein, is the most frequent type of post-translational modification (PTM). More than 50% of human proteins are subjected to the glycosylation machinery for maturation [4]. Protein glycosylation is a conserved kind of PTM that makes indispensable contributions to a wide range of biological processes including protein folding, stability, and interaction with other molecules [5,6,7]. The two major types of protein glycosylation are N- and O-linked glycosylation, which differ in the sugar–protein linkage types. N-glycosylation confers the covalent attachment of an N-acetylglucosamine (GlcNAc) to the nitrogen atom on the side chain of an asparagine (Asn) mediated by an N-glycosidic bond. O-linked glycosylation, on the other hand, occurs most frequently on the oxygen in serine (Ser) or threonine (Thr) residues on substrates. 

Most membrane and secreted proteins, including immune receptors and ligands, are modified by glycosylation. Furthermore, glycan alterations on the cell surface are universally seen in cancers [8]. Immune cells recognize the abnormal glycosylation on cancer cells, and this recognition often leads to an inhibitory immune process. Thus, the glycosylation of immune receptors and ligands plays a vital role in cancer immunity. For example, glycosylation of PD-1, PD-L1, or B7-H4 is required for appropriate ligand–receptor engagement and subsequent function in the antitumor immunity in multiple types of cancers [9,10,11,12,13]. In addition, it has been shown that both co-inhibitory (i.e., PD-L1, PD-1, CTLA-4) and co-stimulatory immune receptors (i.e., 4-1BB, OX40, GITR) are glycosylated, and the glycosylation status and glycan structure of immune receptors vary depending on cell type [13]. 

Here, we will discuss how protein N-glycosylation affects antitumor immunity with a primary pinpoint on its role in the modulation of immune receptors/ligands. Current progress in the development of N-glycosylation targeted immunotherapy will also be highlighted.

## 2. Background of N-Linked Glycosylation

N-linked glycosylation (N-glycosylation), accounting for the modification of over 90% of glycoproteins, is a common and conserved type of PTM in the mammalian system [14]. It should be clarified that the initiation of N-glycosylation takes place during protein synthesis (co-translationally) although glycosylation is commonly categorized as a kind of PTM. The whole process of protein N-glycosylation is achieved in the endoplasmic reticulum (ER) and Golgi by the following four steps: (1) Formation of a dolichol phosphate-bond precursor which contains 14 sugar molecules (Glc_3_-Man_9_-GlcNAc_2_). The mature precursor is located in the ER lumen and serves as the glycan donor for the subsequent steps. (2) Covalent attachment of the glycan to the substrate. The multiunit oligosaccharyltransferase (OST) complex catalyzes the selective transfer of the 14-sugar block to the Asn residues in the consensus glycosylation sequon (Asn-X-Ser/Thr) on a nascent polypeptide that is translocated into the ER lumen. The dolichyl-diphosphooligosaccharide-protein glycosyltransferase subunit (STT3) is known as the active subunit of the eukaryotic OST. Mammalian cells encode two STT3 isoforms, STT3A and STT3B, which coexist but function distinctively. While the STTA3 complex contributes primarily to co-translational N-glycosylation, STT3B is required for both co- and post-translational glycosylation [15,16]. (3) Early processing of glycoprotein in the ER. After conjugation of the 14-sugar precursor to substrates, further trimming of the glycan occurs with an array of ER-resident glycosidases. This step is the quality control step that ensures the correct folding of the newly synthesized glycoprotein. Misfolded proteins will be recognized and eradicated via ER-associated degradation (ERAD) mechanisms to maintain the quality of proteins [17,18]. (4) Glycan maturation in Golgi [5,19]. This is the step responsible for creating an extensive repertoire of glycan structures including high-mannose, hybrid, and complex N-glycans. A series of glycosidases and glycosyltransferases complete the final steps of sugar trimming and addition in the Golgi apparatus. A range of diverse glycan structures can be formed on each given (Asn-X-Ser/Thr) sequon, resulting in great heterogeneity (termed as glycoforms) of glycoproteins and an enormously expanded proteome. At this point, the synthesis of a glycoprotein is accomplished.

## 3. N-Glycosylation in Cancer Immunity

The glycobiology of cancer and the potential value of protein glycosylation as a therapeutic target have been extensively investigated [20,21]. However, the involvement and impact of glycosylation in cancer immunology is still overlooked to some extent. To date, immunotherapy aiming to augment/normalize the host immune response against tumors has evolved as a fundamental strategy in the landscape of cancer remedy [22,23]. The interaction between the host immune system and cancer is tightly regulated through a set of immune stimulatory and inhibitory membrane molecules expressed by immune cells, cancer cells, and other relative stromal cells. In Figure 1, we summarize the immune receptors and cytokine/chemokine receptors that are functionally modulated by N-linked glycosylation. In the following paragraphs, we will discuss how N-glycosylation regulates immune receptors.

### 3.1. N-Glycosylation of Stimulatory Immune Receptors

Cytotoxic T lymphocytes are the primary mediators of the antitumor immune response. T-cell-receptor (TCR) recognition of immunogenic peptides presented by the major histocompatibility complex (MHC) molecule provides the initial activation signal to license the adaptive immune attack against cancer. The TCR is a highly glycosylated multisubunit complex. It has been reported that reduced N-glycan branching on TCRs caused by N-acetylglucosaminyltransferase V (GnT-V/MGAT5) deficiency was associated with hyperimmune diseases in animal models and patients [24], underlining the negative impact of N-glycosylation on TCR function. In agreement with the previous study, Kuball et al. noticed that the selective removal of conserved N-glycans on α and β chains of the TCR enhanced its functional avidity and improved recognition of tumor cells, augmenting the efficacy of TCR-engineered T cells for adoptive transfer treatment [25,26]. 

According to the “two signal model” established decades ago, TCR signaling alone, without co-stimulatory signals, is not sufficient to induce a maximized T-cell response [27]. CD28, a surface glycoprotein expressed on T cells, is one of the best-described immune receptors that modulate immune responses by transmitting secondary signals for T-cell activation upon the interaction with CD80/CD86 (B7-1/B7-2) expressed on antigen-presenting cells (APCs). Ablation of N-glycosylation on CD28 by introducing a point mutation or inhibiting endogenous N-glycosylation enzymes dramatically increased its binding affinity to CD80 and amplified the downstream signal activation, indicative of the negative regulation of CD28 function by N-linked glycosylation [28]. 

In addition to the TCR-mediated primary signal and CD28-mediated secondary signal, a broad spectrum of membrane receptors is involved in fine-tuning the T-cell immune response, many of which are positively or negatively modulated by N-glycosylation, as illustrated in Figure 1A. 4-1BB (CD137/TNFRSF9) is an inducible tumor necrosis factor receptor (TNFR) family co-stimulatory receptor expressed by effector cells including T cells, natural killer (NK) cells, as well as some dendritic cells (DCs). Upon interaction with the trimerized 4-1BB ligand (4-1BBL), cross-linked 4-1BB activates the downstream signal to promote cell expansion and cytokine release [29,30]. As revealed by its crystal structure, the N-glycans on 4-1BB are away from its interface with 4-1BBL, suggesting that N-glycosylation may not be involved in the receptor-ligand interaction of 4-1BB [31,32]. While the engagement of 4-1BB to 4-1BBL was thought to be sufficient for its signal transduction, it was later unveiled by Madireddi et al. that galectin-9 (Gal-9) is necessary for 4-1BB aggregation and signaling by binding to the terminal galactose moieties of N-glycans on 4-1BB. The activation of 4-1BB by 4-1BBL or an agonistic antibody was diminished in Gal-9-deficient mice [33]. As expected, deglycosylated 4-1BB showed markedly decreased binding affinity for Gal-9 [33], which was also validated by structural investigation [31]. 

Natural killer cell receptor 2B4 (CD244) was initially defined as an activating marker on NK cells; however, later studies identified its expression on a variety of immune cells including CD8^+^ T cells, DCs, and myeloid-derived suppressor cells (MDSCs) [34], and it was found to have the potential to mediate both activating and inhibitory signals depending on the adaptor molecules recruited to the cytoplasmic domain of 2B4 [34,35]. The ligand of 2B4 is CD48, which is constitutively expressed by hematopoietic cells [36]. Some of the N-linked sugars on 2B4 are located in close proximity each other within the CD48 binding region and removing 2B4 N-glycosylation by glycosidase digestion or point mutation attenuated its binding to CD48. In addition, ablation of N-glycosylation on NK cells resulted in decreased target cell killing. These findings suggested the requirement of N-glycosylation for the 2B4 activation signal [37]. 

In some cases, N-glycosylation may play bidirectional roles in the regulation of certain receptors such as the inducible T-cell co-stimulator (ICOS/CD278). ICOS belongs to the CD28/B7 superfamily and delivers a co-stimulatory signal that drives T-cell proliferation, cytokine release and cytotoxicity [38,39]. The lack of the N89-linked glycan led to defective membrane localization and cellular function of ICOS [40]. Interestingly, a recently published study found that mutagenesis of N110, one of the three putative glycosylation sites on ICOS, resulted in a 4.3-fold enhancement in binding affinity to its ligand (ICOS-L) [41]. Collectively, N-glycosylation maintains the surface localization ability of ICOS but hampers its ligand binding. 

NK cells mediate cancer cell clearance regardless of antigen-specificity. NKp30 (NCR3/CD337) is a natural cytotoxicity receptor (NCR) that is expressed on NK cells and stimulated by tumor-expressed B7-H6 to elicit a cytolytic effect. It has been shown that human NKp30 is N-glycosylated on three ectodomain sites (N42, N69, and N121). Moreover, B7-H6 binds and activates glycosylated NKp30 preferably in comparison to the deglycosylated mutants (especially N42Q and N69Q) [42]. A more recent study also demonstrated that N-glycosylation is necessary for NKp30 oligomerization which initiates receptor activation [43].

Major histocompatibility class I-related chain molecule A (MICA) can be expressed on tumor cells upon stress and deliver an “eat me” signal recognized by the natural killer group 2D (NKG2D) receptor on NK cells, γδ T cells, and CD8^+^ αβ T cells [44]. Multiple groups showed similar results indicating that MICA cannot be efficiently transported to the cell membrane if N-glycosylation on its extracellular domain is impaired by point mutations or 2-deoxy-glucose (2-DG, a glucose analog that perturbates protein N-glycosylation by blocking the formation of the lipid-linked oligosaccharide precursor [45]) treatment [46,47,48]. 

### 3.2. N-Glycosylation of Inhibitory Immune Receptors

The behaviors of many inhibitory immune receptors are also modulated by N-glycosylation (Figure 1B). PD-1/PD-L1-mediated T-cell exhaustion is probably the most extensively characterized mechanism that induces the dysfunction of antitumor immunity. The primary function of the PD-1/PD-L1 axis is to maintain peripheral tolerance and ensure the desired level of T-cell activation. However, tumors hijack this mechanism to escape from antitumor immunity. Upon the engagement with tumor/APC-associated PD-L1 and PD-L2, PD-1 on T cells form clusters with the TCR complex, recruit the phosphatase Src homology 2 domain-containing tyrosine phosphatase 2 (SHP2) and dephosphorylate the effector proteins, especially CD28, within its proximity. This process, therefore, restrains TCR signaling and balances T-cell immunity and tolerance [49,50]. It has been established that PD-1 requires N-glycan to maintain its abundance and surface localization. Unglycosylated PD-1 is more intensively ubiquitinated followed by rapid degradation [51]. Notably, KLHL22 was identified as an E3 ligase that promotes the ubiquitination of incompletely glycosylated PD-1 and mediates its degradation in the cytoplasm before it is transported to the cell surface [52]. More importantly, glycosylation of PD-1 governs its interaction with its partner PD-L1 [51]. Liu et al. also confirmed that N-glycan is required for the functional binding of certain therapeutic antibodies against PD-1 [53]. 

PD-L1, as the primary binding partner of PD-1, is a heavily glycosylated B7 family protein that is expressed on malignant cells and nucleated cells in the tumor microenvironment (TME) [54]. It has been well-characterized that the N-glycosyl groups on the extracellular domain are critical in maintaining PD-L1 stability [55,56]. β-1,3-N-acetylglucosaminyl transferase (B3GNT3) was identified as a key glycosyltransferase that promotes PD-L1 N-glycosylation. Knocking out *B3gnt3* in mouse breast cancer cells conferred a decrease in tumor PD-L1 expression and, therefore, potentiated tumor rejection [57]. A splicing isoform of the FK506 binding protein 5 (FKBP51) functions as a co-chaperone and promotes PD-L1 expression in gliomas by catalyzing PD-L1 folding and glycosylation [58]. Abnormal N-glycosylation of PD-L1 triggered by the AMP-activated protein kinase (AMPK) in the ER, on the other hand, targets PD-L1 to ERAD for clearance [56,59]. Glycosylation also affects PD-L1 binding to its partners such as its natural ligand PD-1 as well as therapeutic/diagnostic antibodies. Li et al. showed that N-glycosylation is required for PD-L1 to interact with PD-1 and attenuate TCR signaling [60]. Three broadly used therapeutic PD-L1 antibodies, namely avelumab, durvalumab, and atezolizumab, all favor glycosylated PD-L1 over non-glycosylated PD-L1 for binding [60]. Diagnostic antibodies, on the contrary, cannot detect glycosylated PD-L1 efficiently. The study by Lee et al. highlighted that tissue PD-L1 could not be accurately detected by a regular immunohistochemistry (IHC) antibody (clone 28-8) owing to the decreased accessibility of the antibody to heavily glycosylated PD-L1, resulting in considerable false negatives in pathological examinations. The authors modified the PD-L1 detection protocol by pretreating tissue samples with PNGase F, which unmasked PD-L1 for antibody binding. Of note, PD-L1 expression is broadly accepted as a biomarker for clinical benefit from anti-PD-1/PD-L1 treatment, but a considerable group of PD-L1-negative patients respond to PD-1/PD-L1 blockade as well. Thus, this method significantly improves the correlation between tissue PD-L1 expression and clinical efficacy [61].

PD-L1 is not the only inhibitory B7 family protein that is affected by N-glycosylation. B7-H4 (also known as B7S1/B7x/VCTN1) is another B7 family member arising as an immunotherapy target, especially in PD-L1 low tumors [62,63,64]. To date, B7-H4 remains an orphan ligand without a known receptor. The recent work by Song et al. reported that B7-H4 relies on its N-glycosylation to antagonize degradation via the ubiquitin/proteasome pathway [12]. B7-H3 (also known as CD276) is also a targetable T-cell inhibitory receptor that is associated with poor clinical outcomes of cancers [65,66]. High expression of B7-H3 was observed in tumor cell lines and APCs [65]. It has been found that B7-H3 is aberrantly glycosylated on oral cancer cells [66], but the physiological importance of the modification has not been fully elucidated yet. 

Other co-inhibitory immune receptors such as the cytotoxic T lymphocyte antigen 4 (CTLA-4/CD152) are also under regulation by N-glycosylation. CTLA-4 shares about 30% homology with CD28 but binds CD80/CD86 with higher affinity. As the target of the first cancer immunotherapeutic agent (ipilimumab) in clinic, CTLA-4 outcompetes CD28 in binding to CD80/CD86 on APCs, modulating the fine-tuning of TCR signaling and, therefore, preventing autoimmune diseases caused by hyperactivation of T lymphocytes [67,68]. It has been reported that the N-glycan branching of CTLA-4 can be regulated upon TCR signaling and result in elevated surface retention, thereby suppressing T-cell function and driving immune evasion [69]. 

Other than the B7-related immune-modulatory axis mentioned above, a variety of co-inhibitory receptors developed by immune cells, cancer cells, and immune-suppressive myeloid cells can also be governed by N-glycosylation. T-cell immunoglobulin and mucin-domain containing-3 (Tim-3) is a T-cell-exhaustion marker activated by its primary ligand Gal-9, and leads to cell death upon activation. PNGsase F-treated Tim-3 loses the ability to bind Gal-9 [70], which is similar to what was observed with 4-1BB as discussed above. One recent work published by Yang et al. suggested the interesting *in-cis* cross-talk between PD-1 and Tim-3 in a Gal-9-dependent manner. Specifically, PD-1 harnesses Gal-9 as a “bridge” to bind to Tim-3 on exhausted T cells and, consequently, protects PD-1^+^Tim-3^+^ cells from Gal-9-induced apoptosis [71]. N-glycosylation is involved in the assembly of TIM-3/Gal-9/PD-1 lattices. As indicated by biochemical studies, PD-1 relies on its N116-linked glycan to interact with Gal-9 [71]. 

Excessive adenosine in the TME confers a potent immune-suppressive environment. The extracellular adenosine is produced from a sequential nucleotidase reaction mediated by CD39 (ectonucleoside triphosphate diphosphohydrolase 1, E-NTPDase1) and CD73 (ecto-5′-nucleotidase). Specifically, CD39 cleaves ATP to AMP, and CD73 dephosphorylates AMP to adenosine. Activation of adenosine receptors (primarily A2AR) promotes immune evasion by a broad range of mechanisms including the induction of PD-1 and CTLA-4 expression on effector T cells, recruitment of immune-suppressive cells such as regulatory T cells (Treg) and MDSCs, etc. [72,73]. Studies have proven that both CD39 and CD73 are tightly controlled by N-glycosylation. For instance, CD39 with a combinational deletion of several N-glycosylation sites failed to appear on the cell surface [74]. Similarly, biochemical studies characterized that N-glycosylation-deficient CD73 (N311Q/N333Q) has compromised enzymatic activity along with increased Golgi retention [75]. 

Sialic acids are a group of nine-carbon, negatively charged sugar molecules that terminate both N- and O-glycans. Hypersialylation on cancer cells can play a role in establishing an immune suppressive atmosphere by engaging with sialic acid-binding immunoglobulin-like lectins (Siglec) receptors [76,77]. It should be noted that the appearance of sialic acids is not limited to N-linked glycans though. Siglec-15, which was previously characterized as a Siglec member but shares >30% sequence homology with B7 family molecules, was identified from a non-biased functional screen as a cancer stroma and tumor-associated macrophages (TAM)-associated receptor that interacts with its ligand (uncharacterized yet) on the T lymphocyte surface and dampens the T-cell response [78]. Another study by Chen et al. demonstrated that Siglec-15 could be N-glycosylated in a glucose-dependent manner, and this modification is crucial for stabilizing Siglec-15 by preventing its lysosome-dependent degradation [79]. Like Siglec-15, Siglec-10 is also expressed on TAMs and transmits “don’t eat me signals” upon binding to CD24, a cancer-expressed sialoglycoprotein [80]. Antibody-mediated blockade of the CD24–Siglec-10 interaction robustly augments the phagocytosis of CD24^+^ cancer cells by macrophages [80]. Forgione et al. performed a structural study coupled with molecular modeling analysis and uncovered that Siglec-10 favors sialylated complex-type N-glycans on substrates for binding [81].

### 3.3. N-Glycosylation of Cytokine/Chemokine Receptors

Tumor-immunity interaction is also dictated by the cytokine milieu in the TME. Cytokines produced by immune cells, cancer cells, and other stromal cells deliver activation signals through their corresponding receptors to orchestrate the local and distal immune responses, and many receptors, like the majority of membrane proteins, require N-glycosylation for correct localization, function, and turnover (Figure 1C). Transforming growth factor-beta (TGF-β) is believed to be one of the most crucial cytokines shaping the immune-suppressive microenvironment [82,83,84,85]. Co-targeting TGF-β signaling amplifies the response to immune-checkpoint inhibition in vivo [86]. The data by Kim et al. demonstrated that the type II TGF-β receptor (TGF-βRII) without N-glycosylation on two conserved sites was unable to localize to the cell surface and subsequently led to hindered TGF-β signaling [87]. Several other studies focused on the core fucosylation of TGF-βR and found that TGF-βRI and TGF-βRII lacking core fucose were unable to promote epithelial–mesenchymal transition in renal fibrosis [88,89]. Human granulocyte-macrophage colony-stimulating factor (GM-CSF) is the key cytokine responsible for the survival and differentiation of myeloid cells. In the field of immune oncology, GM-CSF mediates antitumoral effects by recruiting APCs to the TME and stimulating antigen presentation. Thus, it has been shown that the local injection of GM-CSF generates a long-lasting tumor rejection effect. The biological functions of GM-CSF are mediated by its engagement with its receptor, which appears as a heterodimer composed of α chain (GM-CSFRα) and a common β chain (GM-CSFRβc). Despite the fact that intact N-glycosylation is not required for membrane targeting, it was discovered that that the loss of N-glycans on any of the three sites on GM-CSFR led to its failure to bind to GM-CSF with high affinity [87]. CD4^+^CD25^+^FoxP3^+^ Tregs are highly abundant in the TME where they mediate immune tolerance. CD25 is referred to as the α chain of the high-affinity IL-2 receptor and has critical roles in Treg development. The depletion of tumor-infiltrated Tregs by CD25 blockade elicits effective tumor growth arrest in in vivo models [90,91]. Chien et al. demonstrated that glucosamine interferes with N-glycosylation and, therefore, impairs the surface retention of CD25 on CD4^+^ T cells and suppresses Treg differentiation [92]. Tumor necrosis factor (TNF) α is a pro-inflammatory cytokine that plays a dual role, and its function in anticancer immunity is still under debate [93]. TNFs are usually considered a T-cell-activation marker. The loss of TNF signaling components has been shown to drive immune evasion from CD8^+^ T cells and NK cell-mediated killing [94]. On the contrary, studies performed on the melanoma model clearly showed that resistance to anti-PD-1 treatment could be unleashed by TNF blockade [95]. It has been reported by Han et al. that TNF receptor 1 (TNFR1) is N-glycosylated on two Asn sites on the extracellular domain (N151/N202). The loss of either one of the N-glycans resulted in decreased TNF binding and restricted NF-κB activation even though the membrane localization of TNFR1 was unaffected [96]. The dependence of N-glycosylation on ligand binding was also observed on the drosophila TNFR homolog [97]. IL-6 is closely associated with cancer immunotherapy-related adverse events, such as the cytokine release syndrome (CRS) induced by chimeric antigen receptor (CAR) T-cell therapy [98]. Administration of the IL-6 receptor blockade antibody, tocilizumab, has been deployed in clinic to prevent life-threatening CRS following CAR-T-cell infusions [99]. IL-6 binds its receptor complex composed of the IL-6 receptor subunit (IL6R) and IL-6 signal transducer glycoprotein 130 (gp130), thereby activating the subsequent signaling cascades. Following the mutation of all nine N-glycosylation sites, most gp130s are unable to transport to the cell surface and, alternatively, enter the proteasome for degradation [100]. An alternative method for IL-6 signaling involved the formation of an IL-6/soluble IL-6R (sIL-6R) complex that subsequently activates downstream gp130 pathways (a process called trans-signaling). However, the cellular source for soluble IL-6 (sIL-6) remained unclear. Riethmueller et al. unveiled one mechanism by which sIL-6 can be generated, which involved a disintegrin and metalloproteinase 17 (ADAM17)-mediated cleavage of membrane-bound IL-6R. It was demonstrated that without the N-glycans surrounding the cleavage site, sIL-6R cannot be successfully released from the cell membrane [101].

Chemokines are another group of secreted proteins that are involved in immunological processes, primarily by facilitating immune cell migration and function [102]. CC-chemokines compose one major family of chemokines which are defined by the two adjacent cysteines near the N-terminus. Within this family of chemokines, CCL19 and CCL21 both induce the trafficking of T cells and DCs by triggering CCR7 signaling. Hauser et al. identified two N-glycosylation sites on CCR7 (N36 and N292) which regulate the binding of CCR7 to CCL19 and CCL21. The removal of N-glycans on CCR7 amplified downstream signaling transduction and cell migration. In addition, glycosylation-defective CCR7 mutants showed markedly reduced endocytosis [103]. CXC-family chemokines differ from the CC-family by having one additional amino acid between the two N-terminus cysteines. CXCR2 is a CXC-chemokine receptor that uses CXCL1/2/3/5/6/7/8 as its functional ligands. Activation of CXCR2 signaling promotes the infiltration of immune suppressive neutrophils and MDSCs. Thus, inhibition of CXCR2 sensitizes cancer cells to anti-PD-1 therapy and enhances long-term survival as demonstrated in murine pancreatic cancer models [104]. As reported, N-glycosylation is required to maintain CXCR2 on the surface of neutrophils [105]. CXCR3 is the receptor for interferon-inducible chemokines such as CXCL9, CXCL20, and CXCL11. CXCR3, which is expressed on activated CD8^+^ T cells and CD4^+^ T helper cells (Th1), plays an essential role in controlling their infiltration into the TME. β-1,4-Galactosyltransferase 1 (β4GalT1) promotes N-glycosylation of CXCR3 at N22 and N32. While N-glycosylation does not affect the membrane expression of CXCR3, it is required to stabilize the binding of CXCL10 and exert its biological functions [106]. 

## 4. N-Glycosylation as a Target for Cancer Immunotherapy

Protein glycosylation has emerged as a promising target for cancer treatment [107]. However, the combination of immunotherapy and glycosylation-targeting has not been considered extensively, and there is no current clinical investigation with a focus in this area. In Table 1 and Figure 2, we summarize the preclinical studies using inhibitors, antibodies, or novel bioengineering agents targeting glycosylation to refine cancer immunotherapy. As described before, PD-1 is heavily glycosylated and relies on N-glycan to bind to PD-L1. Sun et al. developed a monoclonal antibody, STM418, that could specifically recognize the N58-glycan of PD-1, a site known to be critical for PD-1/PD-L1 binding. In vivo administration of STM418 induced a more efficacious antitumor effect compared to nivolumab and pembrolizumab [51]. In addition, mass spectrometry data showed that the majority of N-glycans on PD-1 are core fucosylated [51,108]. Genetic ablation or pharmacologic inhibition (by 2-fluoro-l-fucose) of α-(1,6)-fucosyltransferase (FUT8) was shown to attenuate PD-1 expression by blocking its core fucosylation and, thereby, hinder PD-1 expression and strengthen the T-cell-mediated immune response [108]. One study showed that PD-1 underwent a much higher level of ubiquitination in Fut8^−/−^ murine CD8^+^ T cells compared to wild type, providing insight into the core fucosylation-induced PD-1 stabilization [109]. Attempts have also been made to target PD-L1 glycosylation in cancer immunotherapy. Li et al. developed a monoclonal antibody, STM108, that selectively recognizes fully glycosylated PD-L1 and induces PD-L1 internalization and degradation [59]. Other than blockade antibodies which are commonly utilized against membrane-expressed immune receptors, many small-molecule inhibitors have also been identified to target the PD-1/PD-L1 pathway (reviewed in [110]). Recently, Bristol Myers Squibb disclosed a series of small molecules that disrupt the PD-1/PD-L1 interaction [111,112]. Among them, BMS1166 not only blocks the PD-1/PD-L1 binding on the cell surface but also abrogates the intracellular production of PD-L1 by interfering with the N-glycosylation process. Under-glycosylated cellular PD-L1 failed to exit the ER and, alternatively, entered proteasomal degradation [113].

Poly (ADP-ribose) polymerase (PARP) inhibition has emerged as a promising therapeutic strategy for cancer treatment, but one drawback is that PARP inhibitors promote PD-L1 expression and cause immune suppression. Deglycosylation of PD-L1 by 2-DG rescued olaparib (PARP inhibitor)-induced immune evasion, and the combination of 2-DG and olaparib sensitized tumor cells to T-cell-mediated killing capacity [114]. Later on, Kim et al. found that 2-DG, as a PD-L1 deglycosylation agent, synergizes with the epidermal growth factor receptor (EGFR) inhibitor, gefitinib, in syngeneic breast cancer models [13]. Shi and colleagues integrated previously published clinical studies, functional screens, and omics databases and identified alpha-mannosidase 2 (MAN2A) as a regulator of T-cell dysfunction by controlling N-glycan maturation of PD-L1 in the Golgi apparatus [117]. Swainsonine, a potent inhibitor of MAN2A1, synergized with PD-L1 blockade in syngeneic cancer models that responded poorly to monotherapy [117]. As described above, B7-H4 depends on N-linked carbohydrates on the extracellular domain to resist proteolysis [90]. Pharmacological inhibition of B7-H4 N-glycosylation using the OST inhibitor NGI-1 largely augmented the antitumor immunity in a breast cancer model [12]. Aberrant surface sialic acid is generally found on cancer cells and drives the formation of an immune-suppressive microenvironment [76,118,119]. Desialylation by knocking out UDP-N-acetylglucosamine-2-epimerase (GNE), the key enzyme responsible for sialic acid biosynthesis, or administration of the sialyltransferase inhibitor, Ac_5_3F_ax_Neu5Ac (developed by Rillahan et al. [120]), reinvigorated the antitumor immune response in vitro and in vivo [115,116]. To optimize the specificity of sialylation targeting therapy, Xiao et al. developed a novel enzyme–antibody conjugate T-Sia, which is composed of a recombinant sialidase and the anti-human epidermal growth factor receptor 2 (HER2) monoclonal antibody, trastuzumab. This strategy allowed for the selective removal of the tumor-associated sialic acid on HER2^+^ tumor cells and, therefore, potentiated the tumor cell susceptibility to NK cell-mediated cytotoxicity [118].

Adoptive cell therapy (ACT), which involves the transfer of ex vivo expanded autologous or allogenic immune cells back to cancer patients, showed substantial promise in personalized cancer treatment [121]. The concept of reprogramming N-glycosylation has also been applied to improve existing ACTs, such as chimeric antigen receptor T (CAR-T)-cell therapy. As is known, failure of CAR-T therapy occurs frequently in solid tumors, and one of the underlying mechanisms is PD-1-induced CAR-T-cell exhaustion [122]. Various efforts in targeting PD-1 have been made to prevent CAR-T resistance, such as the combinatorial therapy with PD-1 blockade antibodies [123,124], engineering of CAR-T cells to secrete PD-1- blocking single-chain variable fragments (scFv) [125], or directly knocking out PD-1 on CAR-T cells utilizing CRISPR (clustered regularly interspaced short palindromic repeats) technology [122]. Shi et al. deleted one N-glycan on PD-1 on CAR-T cells by introducing a point mutation on the N74 codon using a base-editing strategy, thereby dampening the surface PD-1 expression and enhancing CAR-T-cell proliferation and cytokine production. Xenograft models subjected to PD-L1 N74-edited CAR-T cells had delayed tumor development and improved overall survival [119].

## 5. Concluding Remarks

The emergence of immunotherapy has shown unprecedented success in cancer therapy. Over the past decades, we have seen continuous growth in knowledge in regard to mechanisms by which tumors escape from immune attack, and many of those discoveries have been translated from research facilities to clinic. Nevertheless, our understanding of tumor–immune interaction is rather superficial. Thus, it remains a central task to explore the mechanism associated with the diverse responses of patients to immunotherapy.

Glycans constitute one of the most fundamental elements of living organisms and are involved in various physiological and pathological processes. Attachment of carbohydrates to substrates is known as glycosylation, which can be observed in the majority of membrane proteins. The contribution of N-glycosylation to the immune system has become clearer now. There is a growing number of functional and therapeutical investigations that focus particularly on the glycosylation of immune receptors instead of bulk protein expression nowadays, as it has become known that most immune receptors are glycoproteins, many of which were covered in this review. However, the enormous structural diversity of the glycome makes it extremely challenging to perform functional studies of protein N-glycosylation. Nonetheless, thanks to the instrumental, methodological, and computational advancements in glycoproteomics in the past years, we now have a much clearer view of the complex repertoire of glycans and glycoproteins from given biological samples [126]. These novel technologies allow researchers to investigate the molecular basis of protein glycosylation under various pathological conditions and support the development of glycomedicine to ameliorate cancer and other diseases.

## Figures and Tables

**Figure 1 cells-10-01100-f001:**
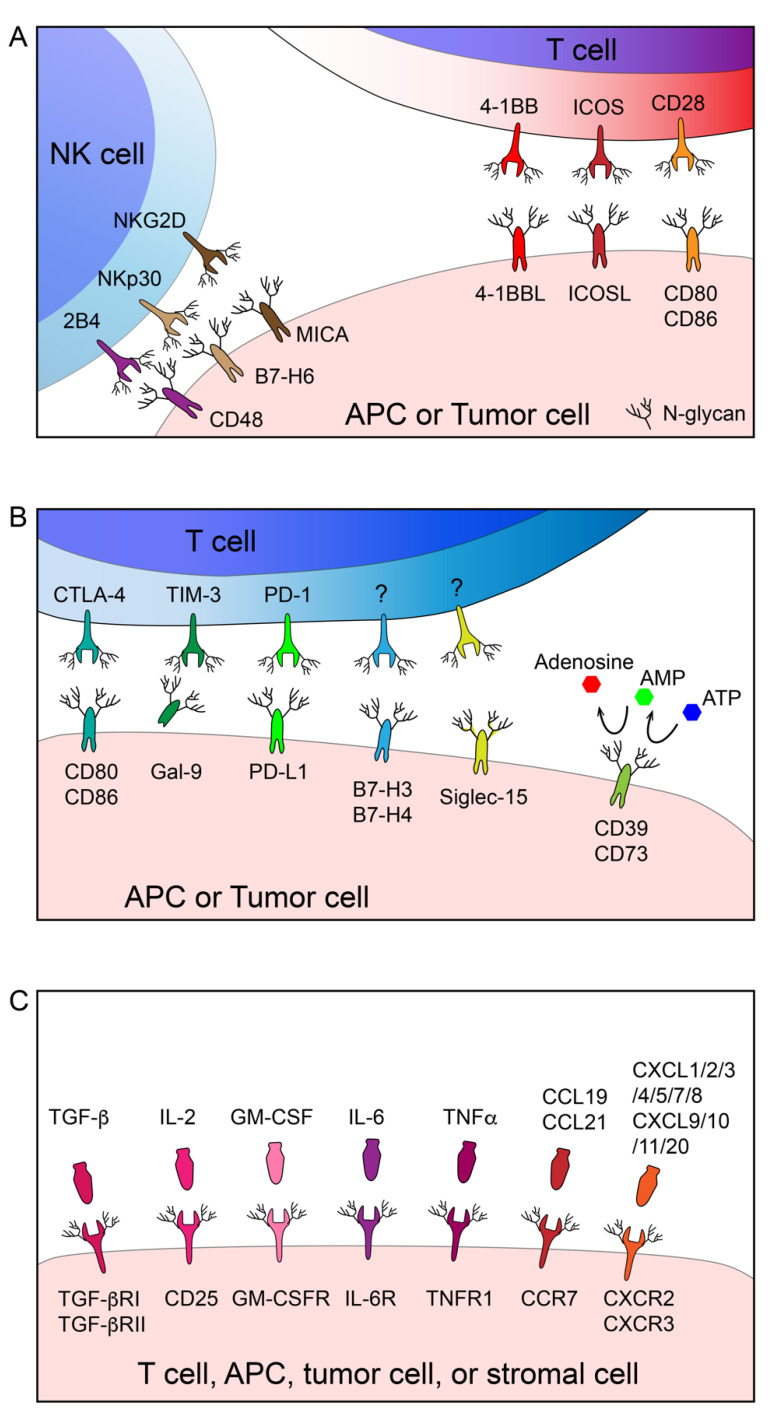
Regulation of the antitumor immune response by N-glycosylation. Many immune cells, APCs, and cancer cell-associated immune stimulatory (**A**), inhibitory (**B**), and cytokine/chemokine (**C**) receptors are functionally modulated by N-linked glycosylation.

**Figure 2 cells-10-01100-f002:**
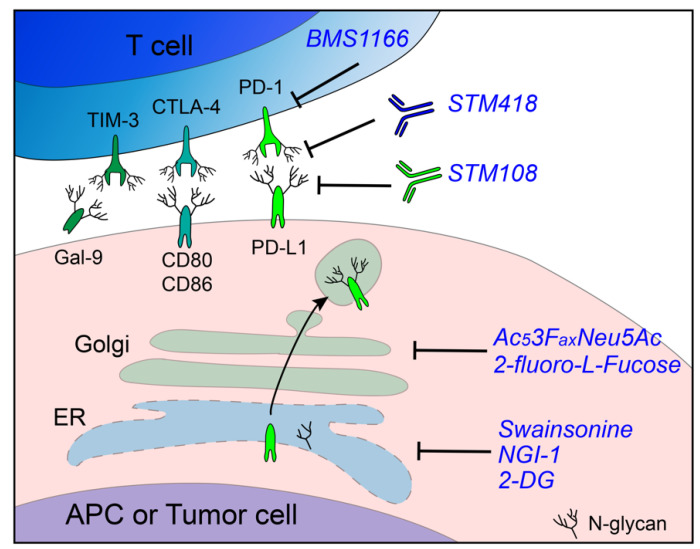
N-Glycosylation as a target for cancer immunotherapy. Remodeling of the immune-suppressive microenvironment can be achieved by targeting receptor N-glycosylation by blocking the catalytic pathway with enzyme inhibitors or developing glycan-specific antibodies.

**Table 1 cells-10-01100-t001:** Strategies to improve cancer immunotherapy by modulating protein N-glycosylation.

Drug name	Type	Target	Reference
2-DG	Sugar analog	Synthesis of lipid-linked oligosaccharides	[13,114]
2-fluoro-l-Fucose	Sugar analog	α-(1,6)-Fucosyltransferase	[108]
Ac_5_3F_ax_Neu5Ac	Sugar analog	Sialyltransferase	[115,116]
Swainsonine	Small molecule	α-Mannosidase II	[117]
NGI-1	Small molecule	STT3A/B	[12]
BMS1166	Small molecule	PD-L1	[113]
STM108	Monoclonal antibody	(N-glycosylated) PD-L1	[59]
STM418	Monoclonal antibody	(N-glycosylated) PD-1	[51]
T-Sia	Enzyme–antibody conjugate	Tumor surface sialic acid	[118]
Base-edited CAR-T	CAR-T	PD-1 (N74)	[119]

## Data Availability

Not applicable.

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
