# Peer review of "Glycosylation of Immune Receptors in Cancer"

_cells, 2021, doi:10.3390/cells10051100_

Round 1

Reviewer 1 Report

The review “Glycosylation of Immune Receptors in Cancer” is really well written and includes the most important aspects. However, I strongly suggest to include figures for all subchapters of chapter 3. This will clarify the relationships between the different aspects.

Author Response

We thank the reviewer for providing the instructive comment. Following the reviewer’s opinion, we added a new figure (Figure 1) which include three panels corresponding to each section of chapter 3. 

Reviewer 2 Report

Well-written and organized review with sufficient bibliographic updates. It does not provide any particular news on this topic, but emphasizes the importance of how protein glycosylation affects the immune response. 

Author Response

We thank the reviewer for the comment.

Reviewer 3 Report

This is a well written paper. I only have minor comments to offer:

  1. Line 39: delete ‘to the’
  2. It would be better to reference Figure 1 earlier (in 3.1 or 3.2) upon first mention of the players in immune checkpoint function. Perhaps also pair the immune checkpoint inhibitors with their ligands clearly. For e.g. CTLA-4 binds CD80/86 but this information is absent from figure. The same for many of the other ligands. It should be clarified which of the receptor(s) has a known ligand and which ones do not have well established relationships. Also, which ones have N-glycan roles and which ones are yet to be established. In general, an entire figure could be dedicated to describing the binding interactions, and the role of N-glycans. This aspect would enhance the impact of the work. The text can be used to expand on the material in the figures by providing more details and context. Currently there is too much information missing in Figure 1.
  3. The inhibitors that have been presented in Figure 1 could be presented in the context of the biosynthetic pathways in section 2 (as a separate figure). Many inhibitors are described targeting Fuc, NeuAc, Glucosidase, OST, Mannosidase etc. Some are early, some are late glycosylation pathway regulators. As written there are observations presented but perhaps this could be added in the context of the biosynthetic pathway. This will simplify the presentation for reader not familiar with glycobiology.
  4. Inclusion of a Table presenting the findings on glycosylation inhibitors would also be helpful, and ex vivo/in vivo studies will be helpful. In this regard, it is important to note that compounds like the Fuc and Neu5Ac analogs are not likely to be clinically viable since they are very non-specific.

Author Response

Response to point #1: Line 39 has been modified as the reviewer indicated. We appreciate the detailed comment.

Response to point #2: We appreciate the reviewer’s suggestion regarding the figure design. The old figure has been replaced with a new multi-panel figure with more details.

Response to point #3: We thank the reviewer for the suggestion. As glycan synthesis requires a large complexity of enzymes with only a few of them being targetable, we decided not to place the compounds into the glycosylation pathway.

Response to point #4: Thanks for pointing that out. We understand that sugar analogs are not specific to glycosylation-related enzymes, therefore we did not emphasize the clinical potential of these agents particularly. Compared to enzyme inhibitors, glycosylation-specific antibodies have better feasibilities in clinical applications.